# Aligned Structured Sparsity Learning for Efficient Image Super-Resolution

Yulun Zhang[1,†]    Huan Wang[1,†,*]    Can Qin[1]    Yun Fu[1,2]
[1]Department of ECE, Northeastern University
[2]Khoury College of Computer Science, Northeastern University

## Abstract

Lightweight image super-resolution (SR) networks have obtained promising results with moderate model size. Many SR methods have focused on designing lightweight architectures, which neglect to further reduce the redundancy of network parameters. On the other hand, model compression techniques, like neural architecture search and knowledge distillation, typically consume considerable memory and computation resources. In contrast, network pruning is a cheap and effective model compression technique. However, it is hard to be applied to SR networks directly, because filter pruning for residual blocks is well-known tricky. To address the above issues, we propose aligned structured sparsity learning (ASSL), which introduces a weight normalization layer and applies $L_2$ regularization to the scale parameters for sparsity. To align the pruned filter locations across different layers, we propose a *sparsity structure alignment* penalty term, which minimizes the norm of soft mask gram matrix. We apply aligned structured sparsity learning strategy to train efficient image SR network, named as ASSLN, with smaller model size and lower computation than state-of-the-art methods. We conduct extensive comparisons with lightweight SR networks. Our ASSLN achieves superior performance gains over recent methods quantitatively and visually.

## 1   Introduction

Image super-resolution (SR) is a fundamental computer vision application, which aims to recover a high-resolution (HR) image from its low-resolution (LR) counterpart. In general, image SR is an ill-posed problem, because there exist many HR candidates for one LR input. To alleviate this problem, more and more researchers have been investigating plenty of deep convolutional neural networks (CNNs) [11, 31, 38] to achieve more accurate mapping from LR image to its HR target.

Deep CNN was firstly introduced for image SR in SRCNN [11] and has attracted continuous attention from both academic and industry communities with its promising SR performance. SRCNN only consists of three convolutional (Conv) layers, hindering its performance. Kim *et al.* achieved notable improvements over SRCNN by increasing the network depth in VDSR [30] with residual learning. Deeper CNNs could be trained successfully with residual blocks [22]. By utilizing simplified residual blocks, Lim *et al.* [38] built a much deeper network EDSR. Zhang *et al.* [64] proposed a residual channel attention network (RCAN), which is one of the deepest SR networks. With increased network size (i.e., deeper and wider), very deep networks, like EDSR [38] and RCAN [64], have achieved remarkable SR performance. However, they also suffer from some drawbacks, such as heavy model parameters, number of operations, and inference time. Therefore, it is impractical to directly deploy them on resource-limited platforms without neural processing units or off-chip memory [36].

---

[†]Equal Contribution
[*]Corresponding author: Huan Wang (wang.huan@northeastern.edu)

35th Conference on Neural Information Processing Systems (NeurIPS 2021).

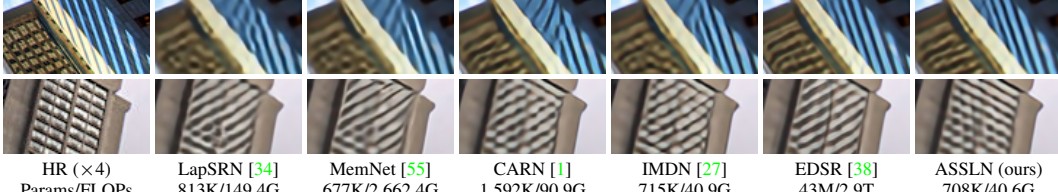

| HR (×4) | LapSRN [34] | MemNet [55] | CARN [1] | IMDN [27] | EDSR [38] | ASSLN (ours) |
| Params/FLOPs | 813K/149.4G | 677K/2,662.4G | 1,592K/90.9G | 715K/40.9G | 43M/2.9T | 708K/40.6G |

Figure 1: Visual results, parameter number, and FLOPs comparison for 4× SR on Urban100 [26] dataset (img_012 and img_020) among lightweight SR networks and a large one EDSR. When calculating FLOPs, we set output size as 3×1280×720. Our ASSL has the smallest number of parameters and FLOPs, while achieving comparable or even better results than others.

From this point of view, more and more works turn to design lightweight network architectures for efficient image SR [1, 27]. Ahn *et al.* proposed cascading residual network (CARN) [1] by implementing a cascading mechanism upon a residual network. Hui *et al.* proposed information multi-distillation network (IMDN) [27]. Lee *et al.* introduced knowledge distillation (KD) [25] for image SR [24, 36] with student and teacher networks. Besides, neural architecture search (NAS) [66, 15] was also utilized for lightweight SR models, like MoreMNAS [8] and FALSR [7]. However, there are still several downsides among these networks: **(1)** The knowledge distillation based methods usually introduce a large teacher network, which will consume more computation resources during distillation training. **(2)** The training in some of these NAS-based methods can also consume heavy computation resources. For example, 8 Tesla V100 GPUs are needed to train a single network for about three days in FALSR [7]. **(3)** Most lightweight SR methods neglect to consider the sparsity or redundancy in the Conv kernels, which can be optimized to be more efficient. In short, more effective, resource-friendly, and general lightweight SR networks are still in need.

To further peel off the redundancy of Conv kernels, neural network pruning techniques [50, 53] are usually introduced to reduce the model complexity. Researchers mainly focus on filter pruning (a.k.a. structured pruning, *e.g.*, [37]) rather than weight-element pruning (a.k.a. unstructured pruning, *e.g.*, [18, 17]) for acceleration. Bridging filter pruning with image SR seems a plausible solution to strike a better performance-complexity trade-off. However, filter pruning methods in image classification can hardly be transferred to SR networks directly. The main reason is that residual blocks have become a essential component in state-of-the-art SR networks to ease the training (*e.g.*, the deep version of EDSR [38] has 80 residual blocks; RCAN [64] even has 200 residual blocks). However, it is well-known that residual connections are hard to prune in structured pruning [37].

To tackle the above issues, we present *aligned structured sparsity learning* (ASSL) for efficient image SR (see Fig. 1). ASSL is essentially a regularization-based filter pruning method. We introduce a weight normalization layer [51] after each convolutional layer and apply sparsity-inducing $L_2$ regularization to the scale parameters in the weight normalization. Besides, a central problem in pruning residual networks in image SR is to align the consequent sparsity structure across different layers (see Fig. 2 constrained Conv layers). In this regard, we propose a novel sparsity structure alignment regularization term to encourage the pruned filter locations across different layers to be the *same*. To the best of our knowledge, our ASSL is the first attempt to leverage filter pruning for efficient image SR. The main contributions of our work can be summarized as follows:

- We propose aligned structured sparsity learning (ASSL) for efficient image super-resolution (SR). To the best of our knowledge, jointly optimizing image SR networks with structured sparsity constraint has received little research attention so far.

- Our ASSL offers a generic framework to structurally prune SR networks with extensive residual connections. To tackle the pruned filter location mismatch issue, a sparsity structure alignment penalty term is introduced to align the pruned filter indices across different layers.

- We employ ASSL to train an efficient aligned structured sparsity learning network (ASSLN), with detailed pruning process visualization for analysis. Our ASSLN achieves superior gains over SOTA lightweight image SR methods quantitatively and visually.

## 2   Related Work

**Deep Image SR Models.** Dong *et al.* [11] firstly introduced CNN with 3 Conv layers for image SR. Residual learning was introduced in VDSR [30], reaching 20 Conv layers. Lim *et al.* [38]

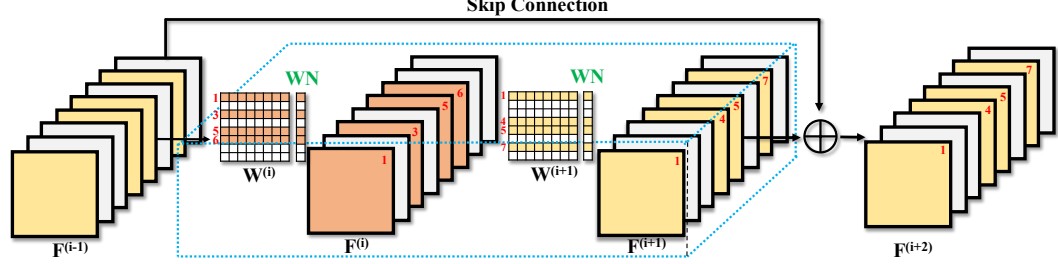

Figure 2: Illustration of filter pruning within a residual block. Feature maps $\mathbf{F}$ are depicted as 3d cubes. Convolutional kernel $\mathbf{W}$ (4d tensor) is expended as a 2d matrix (each row stands for a filter). Both orange and yellow colors mean the *pruned* filters: orange represents the pruned filters in *free Conv layers*; yellow represents the pruned filters in *constrained Conv layers*. We add an extra weight normalization (WN) layer right after each Conv layer. The main point of ASSL is to apply $L_2$ regularization to the unimportant WN scale parameters for sparsity and regularize the indices of pruned WN scales in the constrained Conv layers to be as close as possible to each other.

proposed EDSR with simplified the residual block [22]. Zhang *et al.* [64] proposed an even deeper network RCAN. Liu *et al.* proposed FRANet [39] to make the residual features more focused on critical spatial contents. Later, Zhang *et al.* [65] proposed residual non-local attention for image restoration, including image SR. Mei *et al.* proposed CSNLN [45] by combining feature correlations, and external statistics. Most of them have achieved state-of-the-art results with deeper and wider networks. However, they suffer from huge model sizes and heavy computation operations.

**Lightweight Image SR Models.** Of late, lightweight image SR models have attracted rising attention. Kim *et al.* firstly introduced recursive learning in DRCN to decrease model size [31]. Ahn *et al.* designed a cascading mechanism upon a residual network in CARN [1]. Hui *et al.* proposed a lightweight information multi-distillation network (IMDN) [27]. Meanwhile, neural architecture search was introduced for image SR in FALSR [7]. Besides, model compression techniques, like knowledge distillation, have been investigated for image SR [24]. Lee *et al.* trained a teacher network to distill its knowledge to a student [36]. Although those lightweight networks have achieved great progress, they still need considerable extra computation resources.

**Neural Network Pruning.** Pruning removes redundant parameters in a neural network without performance seriously compromised [50, 53, 5, 6]. Pruning methods can be mainly grouped into structured pruning (*i.e.*, filter pruning) [37, 60, 23, 58] or unstructured pruning [18, 17]. Structured pruning produces *regular* sparsity after pruning, beneficial to acceleration. In contrast, unstructured pruning results in *irregular* sparsity, beneficial to compression (*i.e.*, large sparsity) while hard to leverage for actual acceleration [60, 57]. We focus on *filter pruning* in this work for acceleration.

Most pruning papers focus on finding a better pruning criterion to select insignificant parameters to remove [50, 53, 5, 6, 4]. There are two paradigms to resolve this problem: regularization-based and importance-based. The former selects unimportant weights by adding a sparsity-inducing penalty term, jointly optimized with the original loss function (*e.g.*, [60, 42]). The latter selects unimportant weights via certain derived mathematical formula (*e.g.*, [35, 20, 18, 17, 37, 47, 48]). Note, there is *no* strict boundary between the two paradigms. Several works [10, 57, 58] select unimportant weights by some importance criterion *and* introduce a penalty term for sparsity as well. The proposed method in this paper falls into the last category (see Sec. 3.2 for more details).

To our best knowledge, *no* papers before have successfully joined filter pruning with SR for efficient inference with promising results. We will discuss in length the difficulties within and bridge the gap.

## 3 Proposed Method

We first give a brief view of the image SR problem setting by using deep CNN. We also observe that there exists heavy redundancy in the networks. To pursue more efficient image SR networks, we then propose aligned structured sparsity learning (ASSL) to train lightweight model, resulting in ASSLN.

### 3.1 Deep CNN for Image SR

Given a low-resolution (LR) image $I_{LR}$, image super-resolution (SR) aims to reconstruct its high-resolution (also known as super-resolved) image $I_{SR}$. Such a process can be described as follows,

$$I_{SR} = \mathcal{F}_{SR}(I_{LR}; \Theta), \tag{1}$$

where $\mathcal{F}_{SR}(\cdot)$ is the deep image SR network and $\Theta$ denotes the network parameters. We also model the LR image $I_{LR}$ from its HR counterpart as a degradation process

$$I_{LR} = \mathcal{F}_{\downarrow_s}(I_{HR}), \tag{2}$$

where $\mathcal{F}_{\downarrow_s}(\cdot)$ downscales the original ground truth $I_{HR}$ with scaling factor $s$. The downscaling process may introduce additional noise, blurring, compression, and/or other unknown artifacts. Meanwhile, high-frequency information will be lost, more or less. Image SR models try to recover high-frequency information as much as possible. Here, we focus on efficient neural networks with relatively fewer parameters and computation operations, but comparable or even higher performance.

### 3.2 Aligned Structural Sparsity Learning (ASSL)

In general, our *aligned structural sparsity learning* method is a regularization-based structured pruning method for efficient SR networks. In the following, we will explain (**1**) what parameters are regularized to obtain sparsity, (**2**) how to select unimportant parameters to regularize, (**3**) which is the specific regularization form, and (**4**) how to align the sparsity structure for residual networks.

(**1**) **Regularizing Scales in Weight Normalization**. The goal of structured pruning is to remove filters of a convolutional layer based on some established importance criterion. A natural way is to introduce a gate variable $G$ to control the throughput of each filter (*e.g.*, [40, 33, 29], one filter has a gate accordingly) – zeroed gate implies the associated filter contributes nothing to the subsequent layers, thus can be removed. By regularizing the gate variable, we can know which filters are less important than the others. In classification, previous works [38, 64] have shown regularizing the scaling factor in BN [28] is a natural materialization of this idea. Unfortunately, BN is well-known in practice not useful (even harmful) to SR networks (thus *not* integrated into state-of-the-art SR networks [38, 64]). Therefore, the existing solutions cannot carry over to SR networks.

To resolve this issue, we resort to weight normalization [51] (WN), which proposes to decouple the direction learning of a filter from its norm learning. Specifically, in WN, each filter is normalized to unit length and an extra *learnable* scale parameter is used to learn the filter magnitude,

$$\hat{\mathbf{W}}_i = \frac{\mathbf{W}_i}{||\mathbf{W}_i||_2}, \ \mathbf{W}_i = \boldsymbol{\gamma}_i \hat{\mathbf{W}}_i, \ \text{for} \ i \in \{1, 2, \cdots, N\}, \tag{3}$$

where $\mathbf{W} \in \mathbb{R}^{N \times C \times H \times W}$ represents the 4d convolutional kernel, and $\boldsymbol{\gamma} \in \mathbb{R}^N$ stands for the 1d trainable scale parameters in WN. With weight normalization, we have the $\boldsymbol{\gamma}$ akin to the scale parameters in BN. Then, we can impose certain regularization on $\boldsymbol{\gamma}$ to induce sparsity.

(**2**) **Pruning Scheme and Criterion**. The next question is how to select unimportant $\boldsymbol{\gamma}$ to enforce sparsity (so that we can eventually remove the associated filters). Ideally, we demand a selection mechanism with easy user control. In [41], they sort the BN scales *globally* (namely, scales from *different layers* are compared together). For image SR networks, however, this scheme can hardly work. The main reason lies in the architecture difference of image SR networks vs. the mainstream classification networks. Image SR networks (*e.g.*, RCAN [64]) typically have many more residual connections than those (*e.g.*, ResNet101 [21]) in classification. The global sorting scheme cannot guarantee the two layers that are added together keep the same number of filters. To resolve this problem, we turn to adopt a *local* pruning scheme. For each layer, a pre-defined sparsity level $r$ is given. Filters in a layer are only compared to each other *within that layer*.

As for the criterion issue, previous regularization-based pruning methods typically add a sparsity-inducing penalty term (*e.g.*, $L_1$ regularization [41, 61], $L_2$-norm regularization [60]) to the loss. The advantage of this paradigm is that the network can *learn* to select unimportant filters itself without using a sub-optimal human-defined criterion, yet at a cost – there is no established relation between the penalty strength and the desired sparsity. It is very common in practice we need to hard tune the penalty strength hyper-parameter to strike a good balance between obtaining desired sparsity and not over-penalizing the network [60, 57]. On the other hand, previous pruning works [16, 58] in classification have shown that the simple $L_1$-norm criterion actually works pretty well in practice. $L_1$-norm criterion is well-known only crude in terms of characterizing the incurred loss change when a weight is pruned from the network [35, 20]. However, it is rather simple (no extra acquisition cost during SGD training) with easy user control. Its crudity can also be compensated by the plasticity of deep networks [46, 57]. All taken into consideration, we choose $L_1$-norm as the pruning criterion. Specifically, for $l$-th layer, we sort the filters by their $L_1$-norms, and set those with the least norms as unimportant filters, denoted as set $S^{(l)}$. Then, we apply sparsity-inducing regularization to the

weight normalization scales corresponding to those unimportant filters. Note, we do not enforce any constraint to the important filters since they will stay in the network, no need to restrict their learning.

**(3) Regularization Form**. Here we pin down the sparsity-inducing (SI) regularization form. By conventional wisdom in machine learning , $L_1/L_0$ regularization may be a natural choice for sparsity [13, 3]. However, it is hard to control the proper penalty strength by our observation. Instead, we choose to impose $L_2$ regularization on the scale parameters in weight normalization,

$$\mathcal{L}_{SI} = \alpha \sum_{l=1}^{L} \sum_{i \in S^{(l)}} \boldsymbol{\gamma}_i^2, \tag{4}$$

where $\alpha$ is the scalar loss weight; $\boldsymbol{\gamma}_i$ denotes the $i$-th element of $\boldsymbol{\gamma}$; $S^{(l)}$ represents the unimportant filter index set of $l$-th layer. As inspired by [57, 58], the $L_2$ regularization strength $\alpha$ grows gradually (added by a preset constant $\Delta$ every $T$ iterations) during the sparsity learning process, so that the unimportant filters can be compressed to a negligible amount. As a termination condition, a ceiling limit $\tau$ (a pre-defined constant) is introduced for the regularization co-efficient $\alpha$. When $\alpha$ for unimportant filters reaches $\tau$, the pruning process is finished, followed by finetuning.

The above local pruning scheme can ensure different layers are pruned by the same number of filters. However, it cannot guarantee the pruned *locations* are (close to) the same. This will cause a problem for pruning residual networks about sparsity structure alignment, as explained next.

**(4) Sparsity Structure Alignment**. Residual networks [21] are well-known difficult to prune because the add operations (a.k.a. residual/skip connections) in residual blocks demand the pruned filter indices to be *the same*. Filter pruning via the proposed ASSL method within a residual block is shown in Fig. 2. There are two kinds of convolutional (Conv) layers based on their connection relationship with each other. One group comprises the layers that can be pruned *without any constraint*, which we call as *free Conv layers* in this work; the other consists of layers in which the filters must be pruned *at the same indices*, called *constrained Conv layers*. For a concrete example, in Fig. 2, the layer $\mathbf{W}^{(i)}$ is a free Conv layer and layer $\mathbf{W}^{(i+1)}$ is a constrained Conv layer.

Because of the aforementioned sparsity structure constraint issue, many structured pruning algorithms in classification simply do *not* prune the last Conv layer in a residual blocks [37, 9, 58]. However, this naive solution cannot carry over to the image SR networks. The fundamental reason lies in the architecture difference between SR networks and their counterparts in classification. First, image SR networks typically employ *many more* residual blocks. In some top-performed SR networks (*e.g.*, RCAN [64]), there are even residuals in residuals. Second, each block of SR networks typically has only two Conv layers while ResNets [21] in classification typically have three in a block. Third, the residual block of ResNets [21] typically possesses a bottleneck structure, where the unpruned constrained Conv layer is $1\times1$ Conv, accounting for little FLOPs; while for SR networks, the constrained Conv layers make up an unignorable portion. To see how serious this problem is, taking EDSR as an example, it has 32 residual blocks and each block has two Conv layers. If we do not prune the 2nd Conv layer in a residual block, *half* of the Conv layers are not pruned. In other word, we can only achieve $2\times$ theoretical acceleration *at best*. The real wall-clock speedup probably is even marginal, seriously hindering its practical application.

Given the issue above, it is necessary to prune *all* the layers in residual blocks if we seek acceleration for practical use. Thus, it is straightforward to find a method to align the pruned indices in all constrained Conv layers. Regularization then is a natural choice considering its wide use in enforcing sparsity structure priors in neural network pruning [50, 60, 58].

Concretely, we propose a *sparsity structure alignment* (SSA) regularization term. For two mask vectors $\mathbf{m}^{(i)}, \mathbf{m}^{(j)}$ (for $i$-th and $j$-th constrained layer, respectively) in which zero entries suggest which filters are pruned, if the pruned locations in these two layers are exactly aligned (namely, $\mathbf{m}^{(i)} = \mathbf{m}^{(j)}$), then the inner-product of them, $\mathbf{m}^{(i)} \cdot \mathbf{m}^{(j)}$, is maximized (*e.g.*, Row 2 and 8 in Fig. 3).

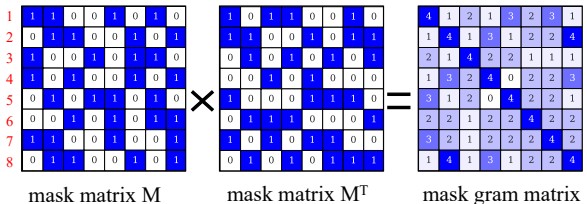

mask matrix M  mask matrix M$^\mathsf{T}$  mask gram matrix

Figure 3: Regularizing the gram matrix of scale matrix.

Therefore, we see that the inner-product of masks is a good optimization target to align the pruned filter locations. For multiple layers, the mask vectors make up a matrix

$M \in \mathbb{R}^{N_c \times N_f}$ (where $N_c$ is the number of constrained layers and $N_f$ is the number of filters in each constrained layer). The inner-products of all combinations make the gram matrix of $M$, $MM^T$. Then the loss term is

$$\mathcal{L}_{SSA} = -\frac{1}{K} \sum_{k=1}^{K} (MM^T)_k, \tag{5}$$

where $K$ is the total number of elements in matrix $MM^T$. One problem of this penalty term is that the 0/1-valued hard mask is not differential. To resolve this, we propose to employ Sigmoid function to obtain *soft* masks. Specifically, given a pre-specified sparsity ratio $r$, we sort the weight normalization scales $\boldsymbol{\gamma}$ in ascending order and obtain the threshold scale as $\boldsymbol{\gamma}_{th}$. Then the soft mask for the $i$-th weight normalization scale in $l$-th layer can be formulated as

$$\mathbf{m}_i^{(l)} = \text{Sigmoid}(\boldsymbol{\gamma}_i^{(l)} - \boldsymbol{\gamma}_{th}^{(l)}). \tag{6}$$

With these soft masks, $MM^T$ become differential and the loss Eq. (5) can be integrated plug-and-play into original SGD optimization (note this penalty term is only imposed on constrained Conv layers).

In the pruning process, this sparsity structure alignment term is jointly optimized with the sparsity inducing loss (Eq. (4)) for a pre-defined number of iterations $t$ (*e.g.*, we set $t=2.56\times10^6$). After that, the sparsity structure is well-aligned and we can apply $L_1$-norm sorting to the scales in weight normalization to decide the unimportant filters in constrained Conv layers.

To sum, the pipeline of the proposed algorithm is: (1) For free Conv layers, we apply sparsity-inducing regularization (Eq. (4)) directly; (2) For constrained Conv layers, we apply sparsity-structure alignment regularization (Eq. (5)) for $N_{SSA}$ (a preset constant) epochs and then apply the sparsity-inducing regularization to them. We provide the detailed algorithm in the supplementary material.

### 3.3 Arm Image SR Models with ASSL

The proposed ASSL approach can be applied as a drop-in module to state-of-the-art SR models – simply add the two penalty terms (Eqs. (4) and (5)) to the original loss function of an SR method. All the features in the original SR method can stay as they are. The proposed penalty term along with weight normalization layers can be implemented very easily on any automatic-differentiation framework for training deep neural networks. When the pruning process is finished, we remove the unimportant filters, which results in a small model. Then we finetune the small model to regain performance following the common practice [50]. Note weight normalization is only needed in the pruning stage. During finetuning, all the weight normalization layers will be removed.

### 3.4 Implementation Details

Here we elaborate the details about how to apply ASSL to constructing lightweight image SR models. First, we revise EDSR baseline (*i.e.*, 16 residual blocks) [38] by removing the final Conv layer to reduce parameters. Same as IMDN [27], the image reconstruction is done via the pixel-shuffle layer [52]. We set kernel size as $3\times3$ for convolution kernel in all convolutional (Conv) layers. For Conv layers with kernel size $3\times3$ (regardless of channel dimensions), zero-padding strategy is used to keep size fixed. We set the initial channel number in the revised EDSR baseline as 256 and then prune it to 48. It should be noted the residual scaling factor in each residual block is set as 1. For $\times2$, we compress the parameter number from 19.5M to 692K and the FLOPs from 4,492.5G to 159.1G.

## 4 Experimental Results

### 4.1 Experimental Settings

**Data and Evaluation.** Following most recent works [56, 38, 63, 19], we use DIV2K [56] and Flickr2K [38] as training data. For testing, we use five standard benchmark datasets: Set5 [2], Set14 [62], B100 [43], Urban100 [26], and Manga109 [44]. The SR results are evaluated with PSNR and SSIM [59] on Y channel of transformed YCbCr space. We also provide model size and FLOPs (a.k.a. Mult-Adds) comparisons. When calculating FLOPs, we set the output size as $3\times1280\times720$.

**Training Settings.** Following [38, 64], we perform data augmentation on the training images, which are randomly rotated by $90°$, $180°$, $270°$ and flipped horizontally. Each training batch consists of 16 LR color patches, whose size is $48\times48$. Our ASSLN model is trained by ADAM optimizer [32] with $\beta_1$=0.9, $\beta_2$=0.999, and $\epsilon$=$10^{-8}$. We set the initial learning rate as $10^{-4}$ and then decrease it to half every $2\times10^5$ iterations. We use PyTorch [49] to implement our models with a Tesla V100 GPU.[*]

---

[*]Our code and trained models are available at https://github.com/MingSun-Tse/ASSL.

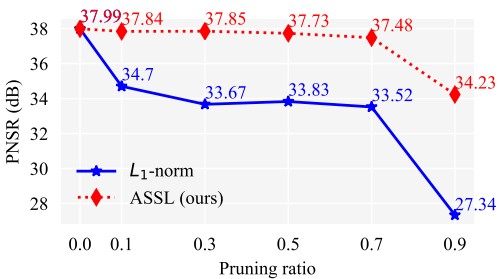 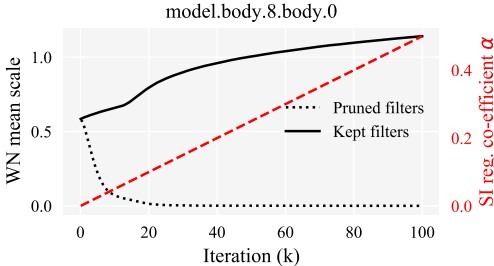

Figure 4: **Left:** PSNR (dB) comparison of models finetuned for only 1 epoch, pruned by $L_1$-norm vs. our ASSL. **Right:** Illustration of the pruning process of Conv layer "model.body.8.body.0" in EDSR. The WN (weight normalization) mean scale of pruned or kept filters are plotted to the left y-axis (in black); regularization multiplier $\alpha$ is plotted to the right y-axis (in red).

## 4.2 Ablation Study

For ablation study, we use EDSR baseline (*i.e.*, 16 residual blocks, 64 features) [38] as backbone, because it is a widely used image SR baseline with public code[†] and results.

**Comparison with Baseline Methods**. We first conduct ablation study to demonstrate the effectiveness of the proposed ASSL method. We compare two baseline approaches here: training from scratch and the $L_1$-norm pruning [37] (which simply removes filters with the smallest $L_1$-norms and is the most prevailing filter pruning method now). The results are presented in Tab. 1. **(1)** The networks pruned by our method

| Pruning ratio | 0.1 | 0.3 | 0.5 | 0.7 | 0.9 |
|---|---|---|---|---|---|
| Params (K) | 1,101.8 | 681.1 | 381.8 | 154.2 | 26.9 |
| FLOPs (G) | 254.5 | 157.7 | 88.9 | 36.5 | 7.3 |
| Scratch | 37.85 | 37.81 | 37.75 | 37.56 | 36.74 |
| $L_1$-norm [37] | 37.91 | 37.81 | 37.73 | 37.58 | 36.87 |
| ASSL (ours) | 37.94 | 37.91 | 37.82 | 37.70 | 37.23 |
| Gain (ours/scr.) | +0.09 | +0.10 | +0.07 | +0.14 | +0.49 |
| Gain (ours/$L_1$) | +0.03 | +0.10 | +0.09 | +0.12 | +0.36 |

Table 1: PSNR (dB) comparison on Set5 ($\times 2$) between ASSL and other two methods to obtain the *same* small network. The unpruned model is EDSR baseline (Params: 1,369.9K, FLOPs: 316.3G, PSNR: 37.99 dB).

*consistently* achieve the best PSNR against different pruning ratios. This shows ASSL is not merely effective (outperforming the scratch training), but also more effective than naively applying the existing pruning method in classification to image SR (outperforming $L_1$-norm pruning). **(2)** Notably, under a *larger* pruning ratio, the advantage of ASSL over scratch training and $L_1$-norm pruning is *more evident* in general, implying that our approach is more effective in extreme pruning cases. **(3)** Another point worth mention is that our method also adopts the $L_1$-norm as pruning criterion, the same as [37]. However, our results are significantly better than theirs. This is because their method does not enforce any regularization to the resulted sparsity structure. Thus the remaining feature map channels are actually *misaligned* in residual blocks of different layers after pruning. Even with a small pruning ratio, the incurred performance damage is very significant, as shown in Fig. 4(Left) – with 0.1 pruning ratio, the network pruned by $L_1$-norm degrades PSNR by 3.29 dB, while ours only decreases PSNR by 0.15 dB. It also indicates that our ASSL maintains most representation ability.

**Regularization Visualization**. To figuratively understand how ASSL works, in Fig. 4(Right) we plot the regularization multiplier $\alpha$ and the mean scale in a weight normalization (WN) layer of EDSR baseline during the ASSL training. The mean scale is split into two parts, pruned and kept. As seen, the regularization multiplier linearly arises against the training epochs as we design. Meanwhile, the mean WN scale of the pruned filters decreases little by little as the penalty becomes stronger. One interesting point is that, note the $L_1$-norms of the mean scale of the kept filters goes up themselves (*no* regularization term is employed to encourage them to grow larger). It means the network *learns to protect itself* from the pruning process, reminiscent of the compensation effect in human brain [14].

## 4.3 Comparisons with Lightweight SR Networks

We compare our lightweight network ASSLN with representative lightweight SR networks: SRCNN [11], FSRCNN [12], VDSR [30], DRCN [31], LapSRN [34], DRRN [54], MemNet [55], CARN [1] and IMDN [27]. We show extensive quantitative comparisons in Tabs. 2, 3 and visual ones in Fig. 5.

**Performance Comparisons.** Tab. 2 shows PSNR/SSIM comparisons for $\times 2$, $\times 3$, and $\times 4$ SR. IMDN [27] ranks the second best except for $\times 4$ SR on Manga109. When compared to all other

[†]https://github.com/sanghyun-son/EDSR-PyTorch

| Method | Scale | Set5 | | Set14 | | B100 | | Urban100 | | Manga109 | |
|---|---|---|---|---|---|---|---|---|---|---|---|
| | | PSNR | SSIM | PSNR | SSIM | PSNR | SSIM | PSNR | SSIM | PSNR | SSIM |
| SRCNN [11] | ×2 | 36.66 | 0.9542 | 32.42 | 0.9063 | 31.36 | 0.8879 | 29.50 | 0.8946 | 35.60 | 0.9663 |
| FSRCNN [12] | ×2 | 37.00 | 0.9558 | 32.63 | 0.9088 | 31.53 | 0.8920 | 29.88 | 0.9020 | 36.67 | 0.9710 |
| VDSR [30] | ×2 | 37.53 | 0.9587 | 33.03 | 0.9124 | 31.90 | 0.8960 | 30.76 | 0.9140 | 37.22 | 0.9750 |
| DRCN [31] | ×2 | 37.63 | 0.9588 | 33.04 | 0.9118 | 31.85 | 0.8942 | 30.75 | 0.9133 | 37.63 | 0.9740 |
| LapSRN [34] | ×2 | 37.52 | 0.9590 | 33.08 | 0.9130 | 31.80 | 0.8950 | 30.41 | 0.9100 | 37.27 | 0.9740 |
| DRRN [54] | ×2 | 37.74 | 0.9591 | 33.23 | 0.9136 | 32.05 | 0.8973 | 31.23 | 0.9188 | 37.92 | 0.9760 |
| MemNet [55] | ×2 | 37.78 | 0.9597 | 33.28 | 0.9142 | 32.08 | 0.8978 | 31.31 | 0.9195 | 37.72 | 0.9740 |
| CARN [1] | ×2 | 37.76 | 0.9590 | 33.52 | 0.9166 | 32.09 | 0.8978 | 31.92 | 0.9256 | 38.36 | 0.9764 |
| IMDN [27] | ×2 | 38.00 | 0.9605 | 33.63 | 0.9177 | 32.19 | 0.8996 | 32.17 | 0.9283 | 38.87 | 0.9773 |
| ASSLN (ours) | ×2 | 38.12 | 0.9608 | 33.77 | 0.9194 | 32.27 | 0.9007 | 32.41 | 0.9309 | 39.12 | 0.9781 |
| SRCNN[11] | ×3 | 32.75 | 0.9090 | 29.28 | 0.8209 | 28.41 | 0.7863 | 26.24 | 0.7989 | 30.48 | 0.9117 |
| FSRCNN [12] | ×3 | 33.16 | 0.9140 | 29.43 | 0.8242 | 28.53 | 0.7910 | 26.43 | 0.8080 | 31.10 | 0.9210 |
| VDSR [30] | ×3 | 33.66 | 0.9213 | 29.77 | 0.8314 | 28.82 | 0.7976 | 27.14 | 0.8279 | 32.01 | 0.9340 |
| DRCN [31] | ×3 | 33.82 | 0.9226 | 29.76 | 0.8311 | 28.80 | 0.7963 | 27.15 | 0.8276 | 32.31 | 0.9360 |
| DRRN [54] | ×3 | 34.03 | 0.9244 | 29.96 | 0.8349 | 28.95 | 0.8004 | 27.53 | 0.8378 | 32.74 | 0.9390 |
| MemNet [55] | ×3 | 34.09 | 0.9248 | 30.00 | 0.8350 | 28.96 | 0.8001 | 27.56 | 0.8376 | 32.51 | 0.9369 |
| CARN [1] | ×3 | 34.29 | 0.9255 | 30.29 | 0.8407 | 29.06 | 0.8034 | 28.06 | 0.8493 | 33.50 | 0.9539 |
| IMDN [27] | ×3 | 34.36 | 0.9270 | 30.32 | 0.8417 | 29.09 | 0.8046 | 28.17 | 0.8519 | 33.61 | 0.9444 |
| ASSLN (ours) | ×3 | 34.51 | 0.9280 | 30.45 | 0.8439 | 29.19 | 0.8069 | 28.35 | 0.8562 | 34.00 | 0.9468 |
| SRCNN[11] | ×4 | 30.48 | 0.8628 | 27.49 | 0.7503 | 26.90 | 0.7101 | 24.52 | 0.7221 | 27.58 | 0.8555 |
| FSRCNN [12] | ×4 | 30.71 | 0.8657 | 27.59 | 0.7535 | 26.98 | 0.7150 | 24.62 | 0.7280 | 27.90 | 0.8610 |
| VDSR [30] | ×4 | 31.35 | 0.8838 | 28.01 | 0.7674 | 27.29 | 0.7251 | 25.18 | 0.7524 | 28.83 | 0.8870 |
| DRCN [31] | ×4 | 31.53 | 0.8854 | 28.02 | 0.7670 | 27.23 | 0.7233 | 25.14 | 0.7510 | 28.98 | 0.8870 |
| LapSRN [34] | ×4 | 31.54 | 0.8850 | 28.19 | 0.7720 | 27.32 | 0.7280 | 25.21 | 0.7560 | 29.09 | 0.8900 |
| DRRN [54] | ×4 | 31.68 | 0.8888 | 28.21 | 0.7720 | 27.38 | 0.7284 | 25.44 | 0.7638 | 29.46 | 0.8960 |
| MemNet [55] | ×4 | 31.74 | 0.8893 | 28.26 | 0.7723 | 27.40 | 0.7281 | 25.50 | 0.7630 | 29.42 | 0.8942 |
| CARN [1] | ×4 | 32.13 | 0.8937 | 28.60 | 0.7806 | 27.58 | 0.7349 | 26.07 | 0.7837 | 30.46 | 0.9083 |
| IMDN [27] | ×4 | 32.21 | 0.8948 | 28.58 | 0.7811 | 27.56 | 0.7353 | 26.04 | 0.7838 | 30.45 | 0.9075 |
| ASSLN (ours) | ×4 | 32.29 | 0.8964 | 28.69 | 0.7844 | 27.66 | 0.7384 | 26.27 | 0.7907 | 30.84 | 0.9119 |

Table 2: PSNR/SSIM comparisons. Best and second best results are colored with red and blue.

| Method | ×2 | | ×3 | | ×4 | |
|---|---|---|---|---|---|---|
| | Params | Mult-Adds | Params | Mult-Adds | Params | Mult-Adds |
| SRCNN [11] | 57K | 52.7G | 57K | 52.7G | 57K | 52.7G |
| FSRCNN [12] | 12K | 6.0G | 12K | 5.0G | 12K | 4.6G |
| VDSR [30] | 665K | 612.6G | 665K | 612.6G | 665K | 612.6G |
| DRCN [31] | 1,774K | 17,974.3G | 1,774K | 17,974.3G | 1,774K | 17,974.3G |
| LapSRN [34] | 813K | 29.9G | N/A | N/A | 813K | 149.4G |
| DRRN [54] | 297K | 6,796.9G | 297K | 6,796.9G | 297K | 6,796.9G |
| MemNet [55] | 677K | 2,662.4G | 677K | 2,662.4G | 677K | 2,662.4G |
| CARN [1] | 1,592K | 222.8G | 1,592K | 118.8G | 1,592K | 90.9G |
| IMDN [27] | 694K | 158.8G | 703K | 71.5G | 715K | 40.9G |
| ASSLN (ours) | 692K | 159.1G | 698K | 71.2G | 708K | 40.6G |

Table 3: Model size and Mult-Adds comparisons of lightweight SR networks with different scales.

methods, our ASSLN performs the best on all the datasets across all scaling factors. Specifically, let's take the challenging ×4 SR as an example. Our ASSLN obtains about 0.23 dB on Urban100 and 0.38 dB on Manga109 PSNR gains over the second best method, respectively. These comparisons show the effectiveness of ASSLN, which learns the aligned structured sparsity. Different from careful network designs as most compared methods have done, we start with the existing EDSR baseline [38] and prune it to a much smaller network. We make better use of the internal sparsity of the network and increase the efficiency of the learned network parameters.

**Model Size and Mult-Adds.** Tab. 3 provides parameter number and Multi-Adds comparison with different scales. Although some previous lightweight SR models (*e.g.*, SRCNN and FSRCNN) cost very small number of parameters and FLOPs, they also have limited performance. Compared with recent popular works (*e.g.*, DRRN, MemNet, CARN, and IMDN), our ASSLN has the least parameter number. We also provide operations number with Mult-Adds. Our ASSLN operates least Mult-Adds than most compared methods except for the FLOPs for ×2. When we consider Tabs. 2, 3 together, we find that our ASSLN achieves a better trade-off between performance and resource consumption. Those comparisons indicate that ASSLN reduces parameters and operations efficiently.

**Visual Comparisons.** We further provide visual comparisons (×4) in Fig. 5 for challenging cases. For example, in img_008, we can observe that most of the compared methods cannot recover structural details with proper directions and/or suffer from blurring artifacts. In contrast, our ASSLN can better alleviate the blurring artifacts and recover more structural details. Similar observations can be found in other cases. These visual comparisons are consistent with the quantitative results, demonstrating the superiority of our method. Our ASSLN learns the aligned structured sparsity from a large network and prunes it to a much smaller one, but still maintains most representation ability.

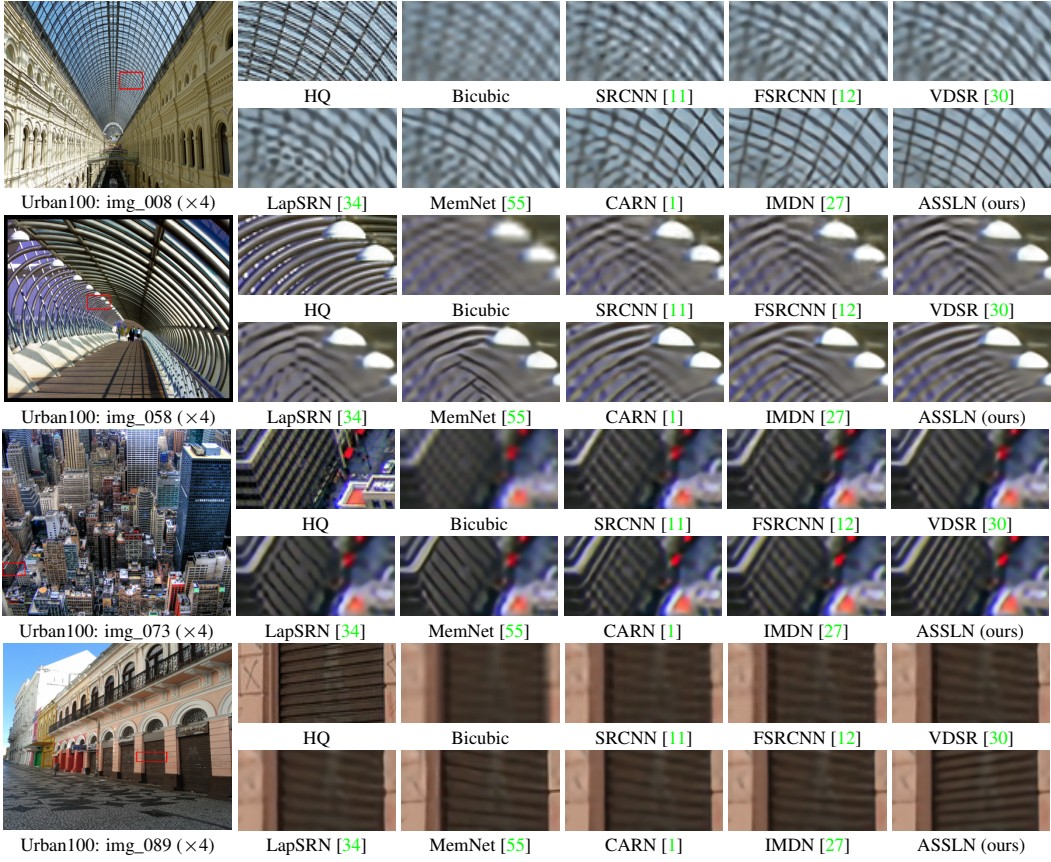

Figure 5: Visual comparison (×4) with lightweight SR networks on Urban100 dataset.

## 4.4 Comparisons with Other Model Compression Methods

To further show the effectiveness of our network pruning method, we compare our ASSLN with representative model compression techniques for image SR. Specifically, we compare with neural architecture search (NAS) based methods (*i.e.*, MoreMNAS-A [8] and FALSR-A [7]) and knowledge distillation (KD) based methods (*i.e.*, CARN+KD [36]). We provide quantitative

| Method | Params | Mult-Adds | Set5 | B100 |
|---|---|---|---|---|
| MoreMNAS-A [8] | 1,039K | 238.6G | 37.63 | 31.95 |
| FALSR-A [7] | 1,021K | 234.7G | 37.82 | 32.12 |
| CARN+KD [36] | 1,592K | 222.8G | 37.82 | 32.08 |
| ASSLN (ours) | 692K | 159.1G | 38.12 | 32.27 |

Table 4: Model size, Mult-Adds, and PSNR comparisons (×2) among model compression methods.

results in Tab. 4. The results of compared methods are copied from their papers directly. Our ASSLN obtains the best performance with the least parameter number and Mult-Adds. With our aligned structured sparsity learning strategy, we do not have to search lots of architectures or train a teacher network, which usually consume plenty of extra computation resources.

## 5 Conclusion

Lately, researchers have been investigating lightweight image super-resolution (SR) networks and achieving promising results with moderate model size and FLOPs. Meanwhile, model compression techniques, like neural architecture search and knowledge distillation, have also been introduced for efficient SR network design. However, they usually consume expensive computation resources. Network pruning is another popular model compression technique, but it is hard to train lightweight SR networks directly because of extensive residual connections in SR. To address these issues, we propose aligned structured sparsity learning (ASSL), which introduces a weight normalization layer and imposes $L_2$ regularization to the scale parameters for sparsity. We further propose a sparsity structure alignment penalty term to align the locations across different layers. We employ such an aligned structured sparsity to train efficient image SR network (ASSLN). Our ASSLN achieves superior performance over recent state-of-the-art methods quantitatively and qualitatively.

**Acknowledgments**. This research is supported by the U.S. Army Research Office Award W911NF-17-1-0367.

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
