# Supplementary Material: Aligned Structured Sparsity Learning for Efficient Image Super-Resolution

**Yulun Zhang**[1,†] **Huan Wang**[1,†,*] **Can Qin**[1] **Yun Fu**[1,2]

[1]Department of ECE, Northeastern University
[2]Khoury College of Computer Science, Northeastern University

## 1 Method

### 1.1 Algorithm

Our proposed aligned structured sparsity learning (ASSL) algorithm is summarized in Algorithm 1.

## 2 Experimental Results

### 2.1 Regularization Visualization

More plots of the pruning process of our ASSL approach on the EDSR_baseline network are shown in Fig. 1. There are in total 16 residual blocks in EDSR_baseline. We evenly pick 5 blocks to plot here (layer name is texted as title in each sub-plot). The free Conv layers are plotted in the left and constrained Conv layers plotted in the right. As seen, similar to layer `model.body.8.body.0` depicted in the main text, all the other plots pose the same trend that *pruned WN scales are compressed towards zero and the kept WN scales arise accordingly to compensate the signal energy loss*. This spontaneous reaction of the kept WN scale parameters prevents the network from catastrophic expressivity damage, hence the performance superiority of our method against the counterparts.

### 2.2 Visual Comparisons

We provide more visual comparisons in Fig. 2. For example, in the right part of img_005, we can observe that most of the compared methods cannot recover structural details with proper directions. In contrast, our ASSLN can better recover more structural details. In img_033, most compared methods suffer from blurring artifacts. While, our ASSLN can better alleviate the blurring artifacts. We can find similar observations in other images. These visual comparisons are consistent with the quantitative results shown in the main paper, demonstrating the superiority of our method. Our ASSLN learns the aligned structured sparsity from a large network and prunes it to a much smaller one, but still maintains most representation ability.

## 3 Explanations for Checklist

### 3.1 Limitations

In this work, we mainly focus on the most commonly used network module: residual block (RB). For other basic modules, like dense connection [4], we have not investigated yet.

---

[†]Equal Contribution
[*]Corresponding author: Huan Wang (wang.huan@northeastern.edu)

35th Conference on Neural Information Processing Systems (NeurIPS 2021), Sydney, Australia.

**Algorithm 1:** Aligned Structural Sparsity Learning (ASSL)

1   **Input**: Pretrained SR network $\Theta$, regularization increment $\Delta$, interval $T$, penalty ceiling limit $\tau$, iterations of SSA $N_{SSA}$.
2   **Output**: Small model $\Theta'$.
3   **Init**: $\alpha = 0$ for all filters; iteration $i = 0$; constrained Conv layer set $\mathcal{C}$.
4   **Init**: Insert a weight normalization layer after each convolutional layer.
5   **Init**: **for** $l = 1 \sim L$ **do**
6      Set $S^{(l)}$ by $L_1$-norm sorting of the filter norms, if $l$ not in $\mathcal{C}$.
7   **end**
8   **Pruning**:
9   **while** *True* **do**
10      **for** $l = 1 \sim L$ **do**
11         **if** $l$ *not in* $\mathcal{C}$ **then**
12            **if** $i\%T = 0$ **then**
13              $\alpha_j^{(l)} = \min(\alpha_j^{(l)} + \Delta, \tau)$ for $j \in S^{(l)}$.
14            **end**
15            Add $\mathcal{L}_{SI}$ (Eq. (4), $\mathcal{L}_{SI} = \alpha \sum_{l=1}^{L} \sum_{i \in S^{(l)}} \gamma_i^2$) to the total loss.
16         **end**
17         **else**
18            **if** $i < N_{SSA}$ **then**
19              Add $\mathcal{L}_{SSA}$ (Eq. (5), $\mathcal{L}_{SSA} = -\frac{1}{K} \sum_{k=1}^{K} (MM^T)_k$) to the total loss.
20            **end**
21            **if** $i = N_{SSA}$ **then**
22              Set $S^{(l)}$ by $L_1$-norm sorting of the filter norms.
23            **end**
24            **if** $i > N_{SSA}$ **then**
25              **if** $i\%T = 0$ **then**
26                 $\alpha_j^{(l)} = \min(\alpha_j^{(l)} + \Delta, \tau)$ for $j \in S^{(l)}$.
27              **end**
28              Add $\mathcal{L}_{SI}$ (Eq. (4), $\mathcal{L}_{SI} = \alpha \sum_{l=1}^{L} \sum_{i \in S^{(l)}} \gamma_i^2$) to the total loss.
29            **end**
30         **end**
31      **end**
32      Loss backward and parameter update by SGD.
33      Iteration adds up: $i \mathrel{+}= 1$.
34      **if** $\alpha$'s *in all layers reach* $\tau$ **then**
35         break. // all layers finished pruning
36      **end**
37   **end**
38   Remove the filters in $S^{(l)}$ for each $l$-th layer. Remove all weight normalization layers (scales merged with filter weights). Rebuild to obtain the pruned model.
39   Finetune the pruned model and output the final model as $\Theta'$.

## 3.2   Potential Negative Societal Impacts

We believe that our efficient image super-resolution (SR) technique: ASSL would benefit to both academic and industry. We think there has few potential negative societal impacts.

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

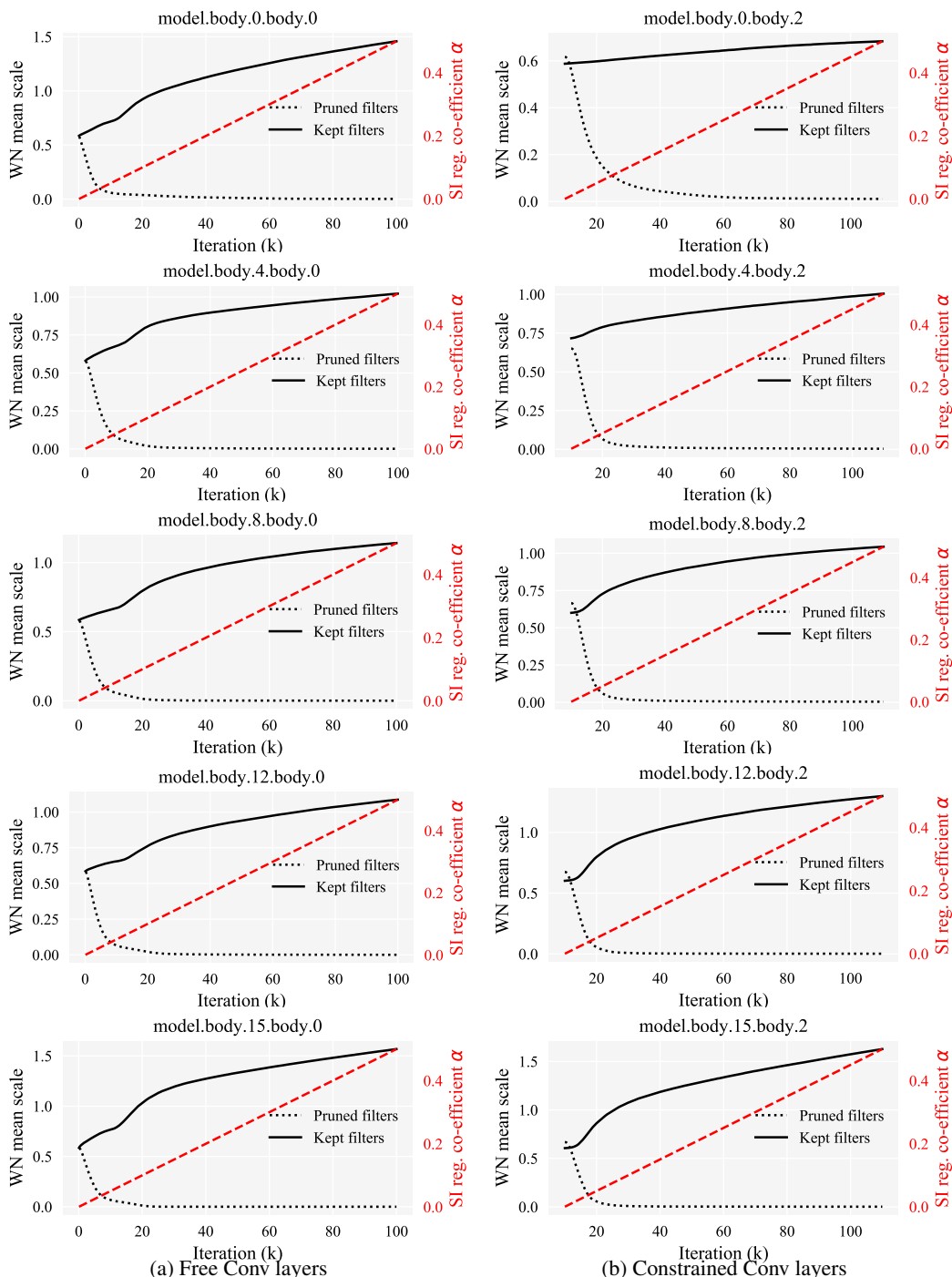

Figure 1: Illustration of the pruning process of five free Conv layers (a) and constrained Conv layers (b) in EDSR_baseline. The weight normalization (WN) mean scale of pruned or kept filters are plotted to the left y-axis (in black); Sparsity-inducing (SI) regularization co-efficient is plotted to the right y-axis (in red). In the main paper, layer "model.body.8.body.0" is presented.

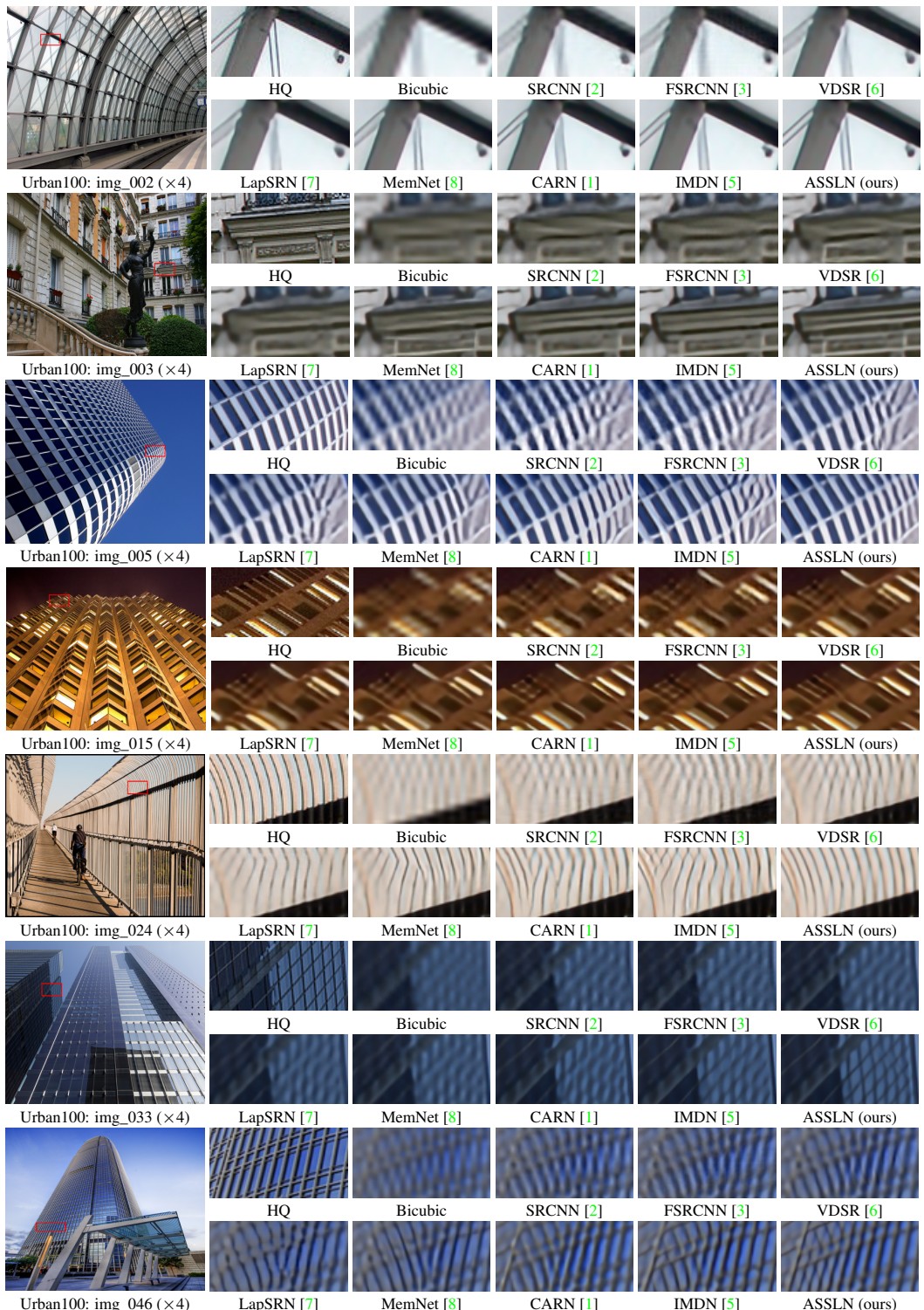

Figure 2: Visual comparison (×4) with lightweight SR networks on Urban100 dataset.