# OpenReview forum: "Aligned Structured Sparsity Learning for Efficient Image Super-Resolution"
_NeurIPS.cc/2021/Conference — NeurIPS 2021 Spotlight_

### Official Review · Reviewer_DsQ3 · 2021-07-09

**Rating:** 8
**Confidence:** 5

**Summary:**

The authors propose a filter pruning method for efficient image super-resolution (SR), named Aligned Structured Sparsity Learning (ASSL). Specifically, they first introduce the Weight Normalization layers into a SR network. Then they propose a regularization term (regularizing the Gram matrix of soft masks of different layers) to penalize the scale parameter in the Weight Normalization layers. The regularization term encourages the consequent sparsity of different layers to present the same structure. Moreover, to maintain expressive power during pruning, they employ a rising L2 regularization scheme to gradually drive the unimportant weights to zero. Experiments show the superiority of the compressed models over those by comparison SR methods.

**Limitations And Societal Impact:**

The authors have adequately addressed the limitations and potential negative societal impact of their work.

**Main Review:**

Pros:
+ This work presents a new filter pruning method (ASSL) based on regularizing the scale parameters in Weight Normalization layers to learn lightweight image SR models.
+ Based on the new method, the authors manage to train a lightweight SR model ASSLN.
+ Strong experimental results -- both quantitative and qualitative results show that the compressed models by the proposed method are obviously better (higher PSNR and sharper details) than the other comparison models.

Cons:
- It would be better to see more recent filter pruning methods compared other than L1-norm pruning [37] in the experiments, such as HRank [1].
- Missing related works: [2, 3] also focus on efficient SR via network pruning. Please discuss how your work is different from theirs.
- The authors mentioned that existing methods for lightweight SR models use heavy computation resources for training. To demonstrate that the proposed ASSL method is more resource-friendly, the authors are supposed to provide more training details, such as the wall-clock training time and specific GPU types.
- Based on the paper, the proposed ASSL method requires a *pretrained* SR model as base model to prune. Although this is very typical in the pruning papers, I am wondering if the method can learn a lightweight SR model *from scratch*.

[1] HRank: Filter Pruning Using High-Rank Feature Map, CVPR, 2020.

[2] DHP: Differentiable Meta Pruning via HyperNetworks, ECCV, 2020.

[3] Efficient Image Super Resolution Via Channel Discriminative Deep Neural Network Pruning, ICASSP, 2020.

------------------------------------------------------------------------------------------------------------------
The author has addressed all my concerns in the rebuttal, and I decide to keep my initial rating and vote for acceptance.

**Time Spent Reviewing:**

15

---

> ### Author Response · Authors · 2021-08-10
> **Author feedback to Reviewer 3 (DsQ3)**
>
> We thank Reviewer 3 (DsQ3) for the valuable comments
>
> `Q3-1:` It would be better to see more recent filter pruning methods compared other than L1-norm pruning [37] in the experiments, such as HRank [1].
>
> `A3-1:`  We compare with HRank in Table 3-1 as follows. The settings of this table are the same as Table 1. As seen, HRank is not as effective as our method.
>
> Table 3-1: Comparisons with HRank for different pruning ratios on Set5 (scale = $\times$2).
>
> | Pruning ratio | 0.1 | 0.3 | 0.5 | 0.7 | 0.9 |
> | :-----: | :----: | :----: | :----: | :----: | :----: |
> | HRank           | 37.91 | 37.83 | 37.74 | 37.60 | 36.89 |
> | ASSL (ours)  | 37.94 | 37.91 | 37.82 | 37.70 | 37.23 |
>
> `Q3-2:` Missing related works: [2, 3] also focus on efficient SR via network pruning. Please discuss how your work is different from theirs.
>
> `A3-2:` Differences from [2]: (1) The main idea of [2] is to employ meta-learning to automatically decide different layer-wise pruning ratios, while our method simply employs the same pre-defined ratio for different layers. (2) DHP learns unimportant filters via regularization, while our method picks them via a predefined importance criterion (i.e., L1-norm). (3) DHP uses L1 regularization to induce sparsity, while our method uses L2 regularization.  Differences from [3]: (1) [3] focuses on selecting unimportant filters by a new criterion Discriminative Information, while we simply adopt L1-norm as criterion. (2) In terms of empirical performance, their method is only evaluated on SRResNet/SRGAN and LapSRN, which are not SOTA SR networks; while our method is shown effective on SOTA SR networks (see Table 2, Table 4, Figure 5).
>
> `Q3-3:` The authors mentioned that existing methods for lightweight SR models use heavy computation resources for training. To demonstrate that the proposed ASSL method is more resource-friendly, the authors are supposed to provide more training details, such as the wall-clock training time and specific GPU types.
>
> `A3-3:` Training the ASSLN model in Table 2 roughly takes 28 hours with 1 NVIDIA XP GPUs, which is significantly cheaper than NAS-based methods like FALSR [7] (it uses 8 Tesla V100 GPUs, 3 days).
>
> `Q3-4:` Based on the paper, the proposed ASSL method requires a pretrained SR model as base model to prune. Although this is very typical in the pruning papers, I am wondering if the method can learn a lightweight SR model from scratch.
>
> `A3-4:` Without a pretrained model, the proposed algorithm can still run. After all, the base model is basically providing better initialization weights. But the performance without using a pretrained base model is worse than that using a pretrained base model by our observation. This coincides with the case in classification. This is also why the current pruning methods typically operate on a pretrained base model.
>
> [1] HRank: Filter Pruning Using High-Rank Feature Map, CVPR, 2020.
>
> [2] DHP: Differentiable Meta Pruning via HyperNetworks, ECCV, 2020.
>
> [3] Efficient Image Super Resolution Via Channel Discriminative Deep Neural Network Pruning, ICASSP, 2020.

---

> > ### Comment · Reviewer_DsQ3 · 2021-08-20
> > **After rebuttal**
> >
> > The author has addressed all my concerns in the rebuttal, and I decide to keep my initial rating and vote for acceptance.

---

### Official Review · Reviewer_Lh3t · 2021-07-09

**Rating:** 8
**Confidence:** 5

**Summary:**

This paper proposes an Aligned Structured Sparsity Learning (ASSL) strategy for SISR. ASSL adds a weight normalization (WN) layer to a SR network and then enforces L2 regularization on the scale parameters in a WN layer. The proposed regularization term can align the final sparsity pattern across different constrained conv layers and deliver considerable speedup without seriously compromising the performance. Experiments show the network trained with ASSL outperforms the other SR networks in terms of performance-cost tradeoff.

**Limitations And Societal Impact:**

The limitations and potential negative societal impact of their work have been discussed in their paper.

**Main Review:**

The overall idea is appealing and the results are promising.

Strengths: (1) The proposed method applies regularization to the scale parameters in the weight normalization (WN) layers. This idea looks novel and logically straight considering that batch normalization (BN) layers are not typically used in SR networks. (2) Regularizing the gram matrix of masks (approximated by sigmoid) to impose sparsity structure is also good. (3) Besides, the empirical performance is strong in either PSNR or visual quality.

Weaknesses: (1) The real acceleration is usually not well-aligned with FLOPs reduction. The authors are encouraged to also report the runtime speedup of the ASSLN. (2) Currently, the paper only uses the same pruning ratio for different layers. This probably is sub-optimal. Is there any idea to do non-uniform pruning? How to decide the proper pruning ratios?

To further improve this submission, I hope the authors could provide sufficient explanations and address my questions.

**Time Spent Reviewing:**

10

---

> ### Author Response · Authors · 2021-08-10
> **Author feedback to Reviewer 2 (Lh3t)**
>
> We thank Reviewer 2 (Lh3t) for the valuable comments
>
> `Q2-1:` The real acceleration is usually not well-aligned with FLOPs reduction. The authors are encouraged to also report the runtime speedup of the ASSLN.
>
> `A2-1:`  We add runtime results in Table 2-1, where the average running time is measured on Urban100 with scale $\times$2. The code and pretrained model are from the official websites. We run the testing on the same machine (3.60 GHz Core(TM) i7-6850K with 64 GB RAM and a Titan Xp GPU).
>
> Table 2-1: Running time comparisons on Urban100 among lightweight SR networks (scale = $\times$2).
>
> | Method | CARN | IMDN | ASSLN (ours) |
> | :-----: | :----: | :----: | :----: |
> | Time (ms)     | 74.3 | 51.8 | 51.0 |
>
> `Q2-2:` Currently, the paper only uses the same pruning ratio for different layers. This probably is sub-optimal. Is there any idea to do non-uniform pruning?
>
> `A2-2:` In pruning, it is broadly believed that layer-wise pruning ratio should reflect the redundancy of each layer (i.e., more redundant layers, larger pruning ratio). However, this idea is hard to materialize since the quantization of redundancy in a neural network is a long-standing hard question per se. So in pruning, there is no standard way to select layer-wise pruning ratios currently. With these said, one possible way is to do layer sensitivity analysis. Namely, tentatively prune each layer by the same amount of ratio and see how the PSNR drops. If the PSNR is sensitive to some layers, we can prune less of them.
>
> Note that, despite using the sub-optimal uniform pruning ratios, our method still achieves the new SOTA performance (see Table 2, Table 4, Figure 5). With better layer pruning ratios, it is supposed to perform even better.
>
> `Q2-3:` How to decide the proper pruning ratios?
>
> `A2-3:` Given a speedup target (or a total FLOPs budget), we choose the pruning ratio simply by trials -- Given a big SR network, pruning it by layer pruning ratio r will get us a smaller network at speedup s. The larger of r, the larger of s. We try small r first and get an s. Increase the r little by little until the speedup s reaches the predefined amount. This process is very easy. Typically, we spend 2 to 3 minutes obtaining the desired ratios.

---

> > ### Comment · Reviewer_Lh3t · 2021-08-22
> > **After rebuttal**
> >
> > The authors have well addressed my concerns, and I've decided to upgrade my rating slightly. I believe this paper presents a contribution worth of acceptance.

---

### Official Review · Reviewer_chSU · 2021-07-11

**Rating:** 8
**Confidence:** 5

**Summary:**

The authors propose a pruning method for training efficient super-resolution (SR) models. It is motivated by the fact that the pruned indices of SR networks should be aligned (due to residual connections) and the batch normalization layers are not commonly employed in top-performing SR models. They propose to apply L2 regularization to introduce weight normalization layers to align the sparsity structure. The proposed method is applied to top-performing SR networks, rendering a new efficient SR model (ASSLN), which is empirically shown to be more favorable than the other compared SR models.

**Limitations And Societal Impact:**

I have checked the limitations and societal impact of their paper. They are properly discussed. I do not have any more concerns in this regard.

**Main Review:**

Pros:

They propose a new pruning method to train efficient networks specifically for SISR. According to the paper, this is “the first attempt to leverage filter pruning for efficient image SR”.

The ideas of regularizing the gram matrices of masks and using the Sigmoid function to obtain soft masks are novel.

Good experimental results demonstrate the effectiveness of the proposed ASSL method.

Cons:

It has been shown in [51] that weight normalization can improve network performance. It is not sure in the paper that the performance superiority of ASSLN is because of the introduced weight normalization layer or the proposed ASSL method. An ablation study is needed to clarify this issue.

The proposed method has several hyper-parameters, such as the $\Delta$ and $T$ (Line 174). Is the proposed ASSL method robust to them? Hyper-parameter sensitivity analysis is needed.

I am a little bit confused with Eq. (6): What is the dimension of $\gamma_{th}$? Is it a vector or scalar?

In all, I like such an idea about network pruning and image SR. The score would be upgraded, if my concerns are well addressed in the rebuttal.


**Time Spent Reviewing:**

2

---

> ### Author Response · Authors · 2021-08-10
> **Author feedback to Reviewer 1 (chSU)**
>
> We thank Reviewer 1 (chSU) for the valuable comments
>
> `Q1-1:` It has been shown in [51] that weight normalization can improve network performance. It is not sure in the paper that the performance superiority of ASSLN is because of the introduced weight normalization layer or the proposed ASSL method. An ablation study is needed to clarify this issue.
>
> `A1-1:` As suggested, we add the ablation study with or without weight normalization as follows in Table 1-1. The settings of this table are the same as Table 1.
>
> Table 1-1: Ablation study in terms of weight normalization and pruning ratio on Set5 (scale = $\times$2).
>
> | Pruning ratio | 0.1 | 0.3 | 0.5 | 0.7 | 0.9 |
> | :-----: | :----: | :----: | :----: | :----: | :----: |
> | Scratch (w/o)     | 37.85 | 37.81 | 37.75 | 37.56 | 36.74 |
> | Scratch (w/)       | 37.84 | 37.81 | 37.77 | 37.57 | 36.73 |
>
>
> As seen, the performance with weight normalization is not statistically better than that without it (so the performance advantage of our method is not attributed to the introduction of weight normalization, but the method itself). This is not surprising considering the task difference between SR and image classification. Especially, consider this -- BN is well-known effective in classification, however, it does not work well in SR networks either.
>
> `Q1-2:` The proposed method has several hyper-parameters, such as the  and  (Line 174). Is the proposed ASSL method robust to them? Hyper-parameter sensitivity analysis is needed.
>
> `A1-2:` The results of varying $\Delta$ and T around their default values are shown Table 1-2 and Table 1-3 below. The pruning ratio here is 0.5. Other settings are the same with Table 1 in the paper.
>
> Table 1-2:  PSNR results of varying $\Delta$ on Set5 (scale = $\times$2).
>
> | $\Delta$ | 1e-5 | 5e-5 | 1e-4 (default) | 5e-4 | 1e-3 |
> | :-----: | :----: | :----: | :----: | :----: | :----: |
> | PSNR    | 37.84 | 37.81 | 37.82 | 37.80 | 37.79 |
>
> Table 1-3:  PSNR results of varying T on Set5 (scale = $\times$2).
>
> | **T**       | **5** | **10** | **20 (default)** | **40** | **80** |
> | :-----: | :----: | :----: | :----: | :----: | :----: |
> | PSNR    | 37.80 | 37.82 | 37.82 | 37.83 | 37.82  |
>
> As seen, the performance is fairly robust to varying $\Delta$ and T -- even the worst PSNR 37.79 is still much better than the comparison method L1-norm (37.73) and Scratch (37.75) in Table 1. Roughly, $\Delta$ is the smaller the better, T is the larger the better, because smaller $\Delta$ (or larger T) implies the pruning process is smoother, beneficial for the network to recover (at the cost of longer training time, though).
>
> `Q1-3:` I am a little bit confused with Eq. (6): What is the dimension of ? Is it a vector or scalar?
>
> `A1-3:` For a conv layer with k filters, $\gamma$ is a vector of k elements. $\gamma_{th}$ is a scalar decided by sorting $\gamma$ (in ascending order) and selecting the number at index int(k*r), where r in the pre-defined sparsity ratio.

---

> > ### Comment · Reviewer_chSU · 2021-08-20
> > **After rebuttal**
> >
> > All my concerns have been well addressed in the rebuttal. The added ablation study with and without weight normalization, and the parameter sensitivity analysis make the proposed method technical reasonable, and better understand. The experimental results also verify the effectiveness and efficiency of the proposed method. Overall, I think this paper has its merits. Therefore, I vote for acceptance.

---

### Decision · Program_Chairs · 2021-09-28

**Decision:**

Accept (Spotlight)

**Comment:**

The paper presents an exciting new approach for image super resolution. All the reviewers are positive about the paper.  Please update the draft with reviewers' suggestions. Congratulations for a nice result!

**Consistency Experiment:**

NeurIPS has a long history of experimentation. In 2014, NeurIPS ran an experiment in which 10% of submissions were reviewed by two independent committees to quantify the randomness in the review process. This year, we repeated a variant of this experiment to see how the quality of the review process has changed over time.  This paper was part of the experiment and was therefore assigned to two committees (consisting of reviewers, an Area Chair, and a Senior Area Chair) that reached independent decisions.  If both committees made the same recommendation, this recommendation was followed. If a single committee recommended acceptance, the paper was accepted (with the exception of a few cases in which the other committee identified what we considered a fatal flaw, e.g., an error in a key result).

This copy’s committee reached the following decision: **Accept (Spotlight)**

The other committee assigned to the paper recommended **Reject**.  You can find the other set of reviews, along with any follow up discussion with the authors here:
https://openreview.net/forum?id=zAuDbrHC6fq